# Long-Term Neurodevelopmental Outcomes of Late Preterm Children: A Pilot Study on the Role of Early Nutrition

**DOI:** 10.3390/nu17223558

**Published:** 2025-11-14

**Authors:** Augusto Biasini, Francesca Agostini, Marcello Stella, Elisa Mariani, Laura Malaigia, Vittoria Rizzo, Fiorella Monti, Erica Neri

**Affiliations:** 1Donor Human Milk Bank Italian Association (AIBLUD), 20126 Milan, Italy; 2Department of Psychology “Renzo Canestrari”, University of Bologna, 40127 Bologna, Italy; f.agostini@unibo.it (F.A.); fiorella.monti@unibo.it (F.M.); erica.neri4@unibo.it (E.N.); 3Pediatric and Neonatal Intensive Care Unit, Maurizio Bufalini Hospital, 47521 Cesena, Italy; marcello.stella@auslromagna.it (M.S.); laura.malaigia@auslromagna.it (L.M.); vittoria.rizzo@auslromagna.it (V.R.); 4Community Pediatric Unit, Ausl Romagna, 47521 Cesena, Italy; elisa.mariani@auslromagna.it

**Keywords:** late preterm, human milk feeding, mixed milk feeding, formula feeding, mild childhood, cognitive development, emotional functioning, behavioral functioning

## Abstract

**Background:** Late preterm delivery accounts for approximately 5% of all births. Although there is a growing body of literature recognizing that children born late preterm are at risk for mild neurodevelopmental issues, these children are often excluded from follow-up programs. From a preventive perspective, it could be useful to promote protective factors for child development. Among these, human milk feeding is highly recommended due to its advantages for child health and development. However, there is limited research on the complex interactions between late preterm birth, human milk feeding, and subsequent cognitive and emotional development. This paper aims to explore these interactions. **Methods**: This study focuses on late preterm children aged 8 to 12 years who are attending school. A trained psychologist conducted assessments of cognitive development using the Wechsler Intelligence Scale for Children-III (WISC-III). Additionally, the family pediatrician gathered neonatal and current growth data, while parents completed the Child Behavior Checklist 6–18 (CBCL) regarding their child’s emotional and behavioral profile. **Results:** A significant impact of early nutrition on children’s cognitive development was observed, with those fed human or formula milk scoring higher than those in the mixed milk group. Additionally, early feeding, high maternal and low paternal perception of externalizing difficulties significantly predicted children’s cognitive development. **Conclusions**: The study suggests the potential role of early nutrition and externalizing difficulties in the cognitive development of late preterm children at school age. Therefore, the inclusion of protective factors such as breastfeeding should be encouraged in future research.

## 1. Introduction

Prematurity represents a severe risk condition for infants and is a leading cause of neonatal and infant mortality [1,2]. In recent decades, the increased survival of preterm infants has been associated with a higher incidence of postnatal consequences, including hospitalizations, chronic illnesses, functional impairments, delays in neurodevelopment, and an overall decline in well-being [3,4,5,6]. Negative consequences could emerge later during middle childhood. The literature has underlined delays in areas such as reading competence, attention, memory, and motor skills, as well as difficulties in emotional and behavioral regulation [7,8], with negative consequences on self-esteem and self-efficacy [9,10,11]. These long-term difficulties could not be detected until they manifested, as screening interventions often did not extend into this age range.

Among all preterm births, late preterm children (LP), born between 34 and 36 weeks of gestation, account for almost 70% of prematurely born infants, with rates varying from 3% to 6% of all births [12]. Although these preterm children were considered at low risk for a long time, there has been growing concern about this population [13,14]. A review by Martínez-Nadal and Bosch [15] highlighted various difficulties in cognitive development, including challenges with executive function, short-term verbal memory, literacy skills, attention, and processing speed. While some of these difficulties are temporary, others can persist and significantly affect academic achievement. However, it is important to note that research on this topic is still limited and is often focused on early childhood.

The growing acknowledgment of the risks associated with late preterm birth has relevant implications for the care of these children. Despite this, the high prevalence of late preterm births has made it difficult for the healthcare system to sustain the financial costs of these programs. Nevertheless, the complications that arise after their manifestation and the expenses related to interventions can further strain resources, making sustainability even more challenging. For this reason, a useful orientation could be represented by the promotion of all the measures aimed at enhancing protective factors starting from their hospitalization in the Neonatal Intensive Care Unit (NICU). Among these, breast milk represents the ideal nourishment to support optimal growth and development, and the benefits on long-term health are widely recognized [16,17,18,19,20].

The literature evidenced relevant benefits for late preterm infants, showing significant differences in body composition in exclusively breastfed infants compared to those fed with formula [21]. Mother’s milk positively correlates with fat-free mass, helping a faster recovery of brain growth and connectome, and the development of an adequate neural network. This benefit could be useful for late preterm infants because they have a quicker tendency to accumulate global adiposity than Very Low Birth Weight infants, especially when fed with formula milk, exposing them to a higher risk of metabolic complications of obesity and cardiovascular diseases at a later age [22].

To our knowledge, little attention has been paid to the long-term effect of the breastfeeding of late preterm children on cognitive outcomes in middle childhood (8–12 years) until now. Understanding the benefits of maternal milk on the later development of late preterm children could, in turn, support the enhancement of breastfeeding practices.

For this reason, we aimed to evaluate the long-term effects of breastfeeding on cognitive and emotional development in late preterm children. Our purpose is to describe and examine the associations among neonatal characteristics, neonatal feeding (maternal, formula, or mixed milk), and cognitive development in late preterm children (8–12 years). Our hypotheses are that late preterm infants fed maternal milk will show a higher level of cognitive development when compared with peers fed with formula or mixed milk. We also aimed to explore whether maternal and paternal perceptions of children could interfere with the associations among variables.

## 2. Materials and Methods

The research is designed as an observational retrospective study on late preterm children.

### 2.1. Sample

Participants of the present study included children born late preterm (34–35 weeks of gestational age) between 2008 and 2010 at the Bufalini Hospital, AuslRomagna (Italy).

According to the study of Huddy et al. [23], we chose to focus on the 34–35 gestational range, excluding those born at 36 GA, given that the last children are at lower risk of developmental delay.

In the current study, children were eligible if they met the following criteria: (a) gestational age (GA) 34–35 weeks; (b) birth weight (BW)  >  1500 g; (c) age range between 8 and 12 years (this range was chosen to ensure that child data would not be influenced by adjustment to transition from pre-school to primary school, and also to prevent possible bias related to physiological changes occurring with the onset of adolescence); (d) exclusive human milk feeding (own mother’s milk or donor milk from the local human milk bank) during NICU stay; (e) absence of neonatal complications (intraventricular hemorrhage grade 3 or 4, periventricular leukomalacia, necrotizing enterocolitis-NEC Bell’s stage 2 during hospitalization). In order to allow the administration of questionnaires, parental fluency in the Italian language was also required for inclusion in the study.

According to the aim of the study, children were included in the following groups: human milk-fed group (HMG), if exclusively breastfed or fed with maternal milk; formula milk group (FMG), if they were fed only formula from birth and did not breastfeed at all; mixed milk group (MMG), for all children with predominant or complementary feeding. Due to the small sample size, we were unable to include a separate group of children fed with prevalent or complementary milk.

A total of 107 children were considered eligible for the study. Among these, 33 children were not assessed due to parental disagreement with participation (*n* = 21), or because we were unable to reach them (*n* = 12). Moreover, 9 participants were excluded from the final sample according to their dropout (*n* = 5) or due to missing data (*n* = 4). The final sample for the study included 62 children, representing 57.94% of all eligible families.

According to the neonatal feeding group, 20 children were included in the human milk-fed group (HMG), 20 in the formula milk group (FMG), and 22 in the mixed milk group (MMG), respectively.

Despite the dropout from the original sample, we considered the sample size adequate for this pilot study, according to power analysis in G*Power 3.1, based on an effect size of f2 = 0.25 and an α = 0.05.

### 2.2. Procedure

The recruitment was allocated in Cesena (Emilia-Romagna, Italy). All family pediatricians of the district (*n* = 20) were contacted and asked about their availability for the recruitment of participants in the study and were trained in the procedure. In the same period, the principal investigator selected the screening cohort of eligible participants from the hospital archive and matched children with the corresponding family pediatrician.

The protocol of assessment included two steps. During the first one, the family pediatricians contacted eligible families by phone, arranging a check-up. In this appointment, the aim of the study was explained to parents, proposing that they participate. In cases of acceptance, written consent to participate was collected, and auxological and clinical variables were assessed.

In the second step, a further appointment was scheduled 15 days after the medical assessment. This assessment took place at the day hospital of the pediatric ward (Bufalini Hospital, Cesena, Italy), where a trained psychologist evaluated the level of child neurodevelopment. While waiting for the child’s assessment, all parents (mothers and fathers) filled out questionnaires on their children’s behavioral and emotional problems in a separate room. In the event of parental absence at the time of assessment, a copy of the questionnaire was provided, requesting their return within a week.

The recruitment took place in October 2019, and the assessment took place until December 2022. Due to the COVID-19 pandemic, the assessment was temporarily paused from March to May 2020 after recruiting 10 participants. All assessments were regularly carried out in person.

### 2.3. Ethics

The study was conducted according to the guidelines of the Declaration of Helsinki and approved by the Institutional Review Board (or Ethics Committee) of CEEIAV (Reg. Sperimentazione n.2034; prot. 2426/2019; 19 June 2019). The study was conducted in accordance with the local legislation and institutional requirements. Written informed consent for participation in this study was provided by the participants’ legal guardians/next of kin.

Participation was completely free of charge and voluntary. Parents might receive a descriptive report on child development and profile as a benefit for their participation in this study.

### 2.4. Measures

#### 2.4.1. Clinical and Demographic Characteristics

Family pediatricians retrieved neonatal variables from hospital clinical records and completed a form including the child’s current clinical variables and parental sociodemographic characteristics at the time of the visit.

#### 2.4.2. Cognitive Development

The level of child development was assessed by the Wechsler Intelligence Scale for Children, Third Edition (WISC-III) [24,25]. WISC-III is an individually administered intelligence test for children aged 6 to 16 years. It includes 13 subtests aimed at measuring intellectual abilities. Specifically, the Verbal domain includes tasks on Information, Similarities, Arithmetic, Vocabulary, Comprehension, and Digit Span. Furthermore, the Performance domain is evaluated with tasks on Picture Completion, Coding, Picture Arrangement, Block Design, Object Assembly, Symbol Search, and Mazes. A trained psychologist tested all children, recording all the tasks completed or failed. All raw scores were computed and converted into final scores. Separate scores are given according to the Verbal Intelligence Quotient (VIQ) and Performance Intelligence Quotient (PIQ): their sum gives a Total IQ score, which represents a general intellectual functioning. Four additional index scores are computed: Verbal Comprehension—VC, Perceptual Organization—PO, Freedom from Distractibility—FD, and Processing Speed—PS. Standardized scores lower than 90 are considered at risk for cognitive delays. All measures were administered in a standard order. WISC is submitted individually in a room in a hospital, in one session of approximately 60 to 120 min, by a trained psychologist blind to the child’s nutritional status. The Italian version of WISC-III has an adequate psychometric reliability, as indicated by the validation study of Orsini and Piconi [25].

#### 2.4.3. Child Psychological Profile

Parents completed the Child Behavior Checklist 6–18 (CBCL) [26,27], a widely recognized parent-reported measure of children’s emotional and behavioral functioning. CBCL includes 113 items regarding the presence and the frequency of specific behavioral and emotional problems. Parents were asked to indicate how accurately each item applied to their child according to a three-point Likert scale (0, not true; 1, somewhat and/or sometimes true; 2, very true or often true). Items are allocated into eight syndrome scales (anxious/depressed, withdrawn, somatic complaints, social problems, thought problems, attention problems, rule-breaking behavior, and aggressive behavior); additionally, the first three syndrome scales (anxious/depressed, withdrawn, somatic complaints) are computed giving scores on internalizing problems, while rule-breaking behavior and aggressive behavior scales contribute to externalizing problems scores. For the present studies, we considered continuous scores at the internalizing and externalizing scales.

A validation study for the Italian version of the CBCL 6–18 by d’Orlando et al. [27] showed good internal consistency (Cronbach’s α > 0.64) and test–retest reliability (Pearson’s r range: 0.57–0.88). In the present study, internal consistency was mostly adequate for all scale scores (Cronbach’s alpha: Maternal CBCL Internalizing symptoms = 0.828; Maternal CBCL Externalizing symptoms = 0.781; Paternal CBCL Internalizing symptoms = 0.834; Maternal CBCL Externalizing symptoms = 0.651) [27].

### 2.5. Statistical Analysis

Preliminarily assumptions for analyses were verified. Descriptive statistics were performed on neonatal variables for the whole sample: specifically, Univariate Analysis of Variance and chi-square tests were used to describe clinical and sociodemographic characteristics of children and their parents.

The effect of the neonatal feeding group on cognitive profile was assessed through a comparison among groups. Specifically, we ran a Multivariate Analysis of Variance for quantitative variables, including as dependent variables WISC mean quotients and indexes and Pearson’s chi-square to explore differences among cut-off scores.

Finally, we used the linear regression model to explore possible predictors of children’s cognitive development. Specifically, we ran separate linear regressions, including in each one as a dependent variable the quotients (Verbal, Performance, and Total) and the index scores (Verbal Comprehension—VC, Perceptual Organization—PO, Freedom from Distractibility—FD, and Processing Speed—PS). As potential predictors, we included the feeding groups and the psychological profile according to maternal and paternal perceptions (Maternal and Paternal CBCL Internalizing and Externalizing scores, respectively). The feeding group is categorized into two variables (“HMG vs. MMG and FMG”; “HMG and MMG vs. FMG”) in order to consider all the comparisons of the three groups. Given that a small number of observations leads to a risk of overfitting the model [28,29,30], we counteracted the negative effects of the small sample size by implementing a bootstrap procedure. We initially performed the analysis in the estimation sample by entering all potential predictors and replicated 2000 times using bootstrap resampling [31]. The final entered model was implemented with WISC scores determined from the bootstrap process.

Data were analyzed using SPSS software version 29.0. A *p*-value < 0.05 was considered statistically significant.

## 3. Results

The clinical and sociodemographic characteristics of the final sample children are shown in Table 1. When we assessed clinical and sociodemographic variables among the feeding group, a global homogeneity emerged (all *p*s > 0.05).

### 3.1. Differences Among Feeding Groups on Cognitive Level of Development

Mean scores at WISC quotients and indexes are shown in Table 2. A global effect of the feeding group emerged in all outcomes, except PS (Table 2). Specifically, Bonferroni post hoc analyses showed that MMG children reported a significantly lower (worse) score than HMG ones in VIQ (*p* = 0.022), PIQ (*p* = 0.008), Total IQ (*p* = 0.006), VC (*p* = 0,05), and PO (*p* = 0.001) scores. The scores of MMG children were also lower (worse) than those of FMG ones for VIQ (*p* = 0.049), Total IQ (*p* = 0.050), and CV (*p* = 0.05) scores. Conversely, no differences emerged between the scores of HMG and FMG children (all *p*s > 0.05).

According to WISC cut-off scores, significant differences emerged among the scores for the Perceptual Organization index. Specifically, the frequency of scores under the clinical cut-off observed in MMG children was significantly higher than the expected one. No significant differences emerged in all comparisons of the other indexes (all *p*s >0.05) (Table 3).

### 3.2. Potential Predictors of WISC Scores

As mentioned above, separate regression analyses were conducted to explore the role of potential predictors of the WISC scores (Table 4).

Overall, the regression analyses showed significant models (all *p*s < 0.05) (Table 4), except for the model for PS (F(5, 54) = 0.399; *p* = 0.848).

It should be noted that we included two potential predictors according to feeding groups, but the factor “HMG vs. MGG and FMG” was excluded in all models because it did not significantly contribute to the models.

Globally, the models showed that parental perceptions of a child’s externalizing problems significantly predicted the child’s WISC scores. Interestingly, the higher the maternal perceptions of problems, the lower the IQ scores; conversely, for fathers’ CBCL scores, associations were in the opposite direction, with a higher IQ for children with elevated externalizing scores (Table 4). Conversely, no significant association emerged for the perceptions of internalizing problems in all models.

When the feeding group’s effect was considered, a significant effect emerged only for the Perceptual Organization index, where formula feeding predicted lower scores for the index (Table 4).

## 4. Discussion

This paper focuses on the assessment of cognitive development in late preterm children at school age. Given that the literature recognized how these children are at risk for difficulties and developmental delays [15,32,33], the identification of protective factors could have clinical and preventive relevance. Specifically, we focused on early nutrition, a factor that is still little studied in the case of late preterm children.

The research is presented as a pilot study, and the results should be considered as preliminary. Therefore, they are discussed in accordance with our hypothesis, taking into consideration the broader literature and the implication for future investigations.

In line with our hypothesis, the results suggested significant differences in the level of cognitive development among the three groups. In particular, children fed with human milk from our sample had the highest quotients and index mean scores. This result could be considered consistent with the previous literature that emphasizes the role early nutrition could play in the later development of full-term [19,34,35] or preterm children [33,34,36,37].

However, in our study, the benefit of human milk emerged only in comparison with children fed mixed milk, with scores similar to those of the formula group. In all comparisons, we observed the lowest scores for the children of the mixed milk group.

It should be noted that these differences mainly emerged when continuous scores are considered. When we analyzed the cut-off score results, a high frequency of risk scores was observed in all groups (i.e., almost 30% of the Total Quotient). The only exceptions were regarding the Perceptual Organization index mean scores, where a delay was observed for MMG. This index investigated the nonverbal fluid reasoning, visual–motor integration, and visual–spatial problem-solving [24], so it represents a strictly cognitive skill.

The specific difficulties observed in the mixed milk group are an unexpected result, and a replication on a wider sample is needed for a confirmation of its plausibility. We considered potential factors that may have contributed to our results. First, when late preterm children are fed mixed milk, different practices could be used, influencing the results. Indeed, no clear definition of mixed milk exists in the literature, as it includes a variety of practices: mothers may adapt their MMG patterns from day to day or month to month, and different mothers follow different practices [38]. Moreover, we could hypothesize that maternal factors associated with mixed milk nutrition practice may have affected the results. Mothers experienced a high level of distress when difficulties in breastfeeding arose, and they could report a high level of guilt [39], especially considering the social stigma surrounding “natural” nutrition [40]. In the case of MMG, we could suppose that the absence of clear guidelines for formula feeding could enhance the difficulties and uncertainties of these mothers, with implications for the infant and their relationship. Given the small sample size recruited for the present study, we were not able to control for variables associated with these practices, but future investigations should consider them to better understand these influences.

Despite the careful consideration of the results, we think that special attention should be given to children fed with mixed milk, including their assessment in all studies aimed at investigating the role of early nutrition in child outcomes. Indeed, the literature states that a substantial proportion of late preterm infants are fed with mixed milk, which could reach almost 30% of late preterm infants from discharge to the weaning period [41,42,43,44].

We also aimed to evaluate the potential predictors of child WISC scores, including both the feeding groups and the parental perceptions of the children in the model. Although our investigation is exploratory and the results should be considered preliminary, this analysis could provide a wider perspective on the child’s condition. First, the results suggest a limited influence of early nutrition, with only a significant effect on the Perceptual Organization index, indicating potential difficulties for formula-fed children. The limit given by the sample size could have contributed to the modest result: although we corrected the regression analyses with bootstrap resampling, further analyses are mandatory to confirm if the benefit of human milk is limited to the early years of life, or if a weak but potential effect could persist, but is not detectable, for small groups.

Another possible consideration regards the potential role that other factors may have had on the actual level of development, directly or in interaction with the early nutrition characteristics. For example, in recent years, accumulating evidence has underscored the critical role of the gut microbiota in early brain development through the gut–brain axis. The microbial colonization of preterm babies is frequently delayed or impaired by cesarean delivery, antibiotic exposure, and formula feeding, all of which contribute to gut dysbiosis during this critical period; however, breastfeeding could play a pivotal role in promoting the colonization of beneficial gut bacteria in preterm infants, particularly strains of Bifidobacterium and Lactobacillus, because these microbes generate short-chain fatty acid (SCFA) [45,46]. To our knowledge, all studies investigating microbial colonization focused on severe preterm babies, while research on late preterm infants is lacking. We believe that it is reasonably plausible that late preterm infants are also affected by the same problem and hope that further studies will include their assessment in the investigation.

The regression analyses also suggested another interesting potential predictor. The results showed the significant effect of CBCL scores. In all models, both maternal and paternal perceptions of the externalizing profile significantly predicted children’s cognitive development, suggesting a possible role of behavioral regulation. The previous literature has confirmed that late preterm children could manifest difficulties in both behavioral regulation and cognitive development, and that these factors could influence each other [47,48]. The present study did not allow for an inference of a causal association due to the small sample size and the indirect assessment of the child’s psychological profile (evaluated by parents). However, given that the results seemed to emerge specifically for externalizing difficulties, we could suppose that these behaviors could be particularly relevant for parents, supporting the need for their investigation.

Finally, the complexity of the results in the present study is also given by the opposite perceptions observed in mothers and fathers: while maternal high CBCL scores (corresponding to severe difficulties in externalizing regulation) are associated with lower WISC scores, in the case of fathers, the relationship was inverse (the higher the regulation difficulties, the higher the WISC IQs/indexes). This unexpected result could also suggest a different sensitivity of the parents to catching children’s psychological profiles and difficulties. Given the small sample size of the study, further analyses were not available, nor were potentially different interpretations of the test by fathers and mothers. However, the results may suggest a complex interaction among parental roles, children’s perceptions, and cognitive development. Further studies are needed to understand if some mechanisms could moderate these associations. We should remember that late preterm children and parents recruited in the present study were not previously included in follow-up programs for severe preterm infants (i.e., VLBW and ELBW). So, it could be useful to understand if the presence of assessment in the previous stage of development could help parents to have a more realistic perception of children’s vulnerabilities and resources.

This result also suggests the relevance of the inclusion of parents in the assessment of the profile of late preterm children, providing a description of difficulties and resources in different contexts. For these reasons, we think that CBCL scores could be very informative for both child screening and suggestions for intervention. Furthermore, the inclusion of both parents (mothers and fathers) is relevant considering the growing interest shown in the literature in the paternal role and its influence on child behavior and development [49,50].

### Strengths and Limitations

The present pilot study has the strength of focusing on a little-studied topic, i.e., the role of early nutrition on the cognitive development at school age of late preterm children. Specifically, it focuses on late preterm children, who are often understudied compared to their more severe preterm peers. Furthermore, the inclusion of a global assessment, including both a professional and parental assessment, could increase the knowledge of the complex profile of these children.

Nevertheless, the many limitations of this study should be acknowledged. First, the results need to be confirmed on larger samples. The recruitment of participants from children born in a single Hospital (Bufalini Hospital in Cesena, Italy) also influenced the small sample size, reducing the power of the analyses which did not allow for the detection of small significant effects or the testing of more sophisticated hypothesis and analyses. A control group of full-term children is also missing. For this reason, the results should be considered preliminary and useful for providing more foundations for hypothesis generation for future phases of research.

Second, we collected data only at one point of observation. In the absence of an assessment in an intermediate step, we cannot exclude the presence of possible factors that emerged in previous ages that could influence the outcomes.

Furthermore, in the present study, the potential influence of confounding characteristics is not investigated. So, the assessment of nutritional variables at birth (i.e., the length and continuity of breastfeeding, the infant age at weaning, delivery method, antibiotic therapy, probiotic supplementation, etc.) and at the time of assessment (type of diet, current eating habits, etc.), as well as additional information on sociodemographic status, are missing. Furthermore the COVID-19 lockdown overlapped the time of the assessment: despite we found no differences between the dropout and final sample, we are aware that the peculiarity of this time of assessment could have influenced the quality our results, and not only the rate of participation. Therefore, all these variables should be included in future investigations.

Finally, although we considered different figures in the assessment of children (pediatricians, psychologists, parents), other relevant informants are missing. In particular, for privacy issues, we could not administer the CBCL to child teachers. Given that children interact daily at school with their teachers and that some emotional and behavioral attitudes could not be shown at home, we recognize that our data could be partial. Additionally, a questionnaire or interview administered directly to children could provide relevant information on their self-perception.

Despite being preliminary, these findings might contribute to the understanding of the implications of early feeding as a protective factor for the cognitive development of late preterm children and the need for the care of these children and their families.

## 5. Conclusions

The present pilot study aimed to evaluate the long-term effects of breastfeeding on the cognitive and emotional development of late preterm children. Considering the high prevalence of late preterm births and the rate of difficulties of these children during primary school, as well as the impossibility of including them in the same follow-up programs as more severe premature infants, the identification of protective factors should represent a priority of the research. This study supports the feasibility of this kind of investigation and encourages the replication on a wider sample.

## Figures and Tables

**Table 1 nutrients-17-03558-t001:** Characteristics of the sample according to group.

	HMG (*n* = 20)	MMG (*n* = 22)	FMG (*n* = 20)	F/X2	*p*
Neonatal Variables					
Birth weight, grams, mean (SD)	2569.25 (414.04)	2023.64 (350.24)	2399.25 (197.98)	14.804	<0.0005
Gestational age, weeks, mean (SD)	34.70 (0.47)	34.59 (0.50)	34.69 (0.48)	0.336	0.716
Length, cm, mean (SD)	45.89 (5.49)	45.85 (2.41)	46.19 (1.44)	0.036	0.965
Cranial circumference, cm, mean (SD)	34.66 (6.79)	31.84 (1.07)	32.13 (1.41)	1.780	0.187
APGAR, mean (SD)	9.60 (0.60)	8.58 (1.47)	9.17 (0.99)	4.451	0.016
Hospitalization, *n* (%)				4.127	0.127
Yes	5 (26.3)	10 (52.6)	5 (25.0)		
No	14 (73.7)	9 (47.4)	15 (75.0)		
Gender, *n* (%)				0.544	0.762
Male	12 (60.0)	11 (50.0)	10 (50.0)		
Female	8 (40.0)	11 (50.0)	10 (50.0)		
Current Variables					
Age, years, mean (SD)	10.25 (0.78)	10.18 (0.91)	10.45 (1.19)	0.422	0.657
Weight, kg, mean (SD)	41.58 (14.78)	35.21 (12.01)	37.69 (8.04)	1.449	0.243
Height, cc, mean (SD)	144.20 (8.59)	138.90 (9.70)	140.77 (9.32)	1.705	0.191
CC, cm, mean (SD)	54.41 (1.79)	53.62 (2.13)	52.48 (5.90)	1.163	0.321
Metabolic disease/condition, *n* (%)	1 (5.0)	3 (13.6)	0 (0.0)	3.331	0.189
Mothers					
Age, years, mean (SD)	43.70 (4.70)	44.35 (4.61)	44.90 (2.51)	0.461	0.633
Education, %				1.209	0.877
Primary/Secondary school	35%	30%	30%		
High school	45%	55%	60%		
University	20%	15%	10%		
Civil status, %				5.967	0.202
Married	90%	60%	80%		
Cohabiting	5%	20%	15%		
Separated/Divorced/Widowed	5%	20%	5%		
CBCL Internalizing symptoms, mean (SD)	49.85 (12.34)	53.14 (8.76)	54.05 (12.37)	0.781	0.462
CBCL Externalizing symptoms, mean (SD)	44.46 (6.76)	46.55 (9.79)	44.80 (6.04)	0.442	0.645
Fathers					
Age, years, mean (SD)	46.60 (6.33)	45.65 (5.00)	47.10 (5.08)	0.3580	0.701
Education, %				5.345	0.254
Primary/Secondary school	20%	30%	50%		
High school	60%	60%	45%		
University	20%	10%	10%		
CBCL Internalizing symptoms, mean (SD)	47.95 (13.12)	45.23 (9.25)	49.35 (12.25)	0.692	0.504
CBCL Externalizing symptoms, mean (SD)	40.10 (7.21)	35.77 (10.78)	38.20 (9.00)	1.176	0.316

NOTE. HMG = human milk group; MMG = mixed milk group; FMG = formula-fed group. CBCL = Child Behavior Checklist 6–18 [26,27].

**Table 2 nutrients-17-03558-t002:** Cognitive development mean scores according to the neonatal feeding group.

	HMG (*n* = 20)	MMG (*n* = 22)	FMG (*n* = 20)	Total Sample (*n* = 62)	F	*p*	η^2^
Verbal IQ-VIQ	103.45 (14.70)	89.64 (18.88)	101.95 (13.98)	98.06 (17.06)	4.712	0.013 ^ab^	0.138
Performance IQ-PIQ	106.10 (10.67)	94.77 (11.11)	101.10 (13.18)	100.47 (12.42)	4.968	0.010 ^a^	0.144
Total IQ	105.05 (13.26)	91.14 (14.77)	101.70 (13.79)	99.03 (15.03)	5.724	0.005 ^ab^	0.162
VC	102.45 (13.87)	90.59 (18.83)	102.45 (13.40)	98.24 (16.45)	4.060	0.022 ^ab^	0.121
PO	110.15 (12.25)	95.91 (12.74)	103.85 (12.39)	103.06 (13.62)	6.891	0.002 ^a^	0.189
FD	104.30 (17.74)	90.91 (12.83)	101.45 (19.45)	98.63 (17.53)	3.750	0.029 ^a^	0.113
PS	100.30 (10.50)	93.18 (15.05)	100.70 (12.45)	97.90 (13.15)	2.294	0.110	0.072

NOTE. Values are mean (sd). HMG = human milk group; MMG = mixed milk group; FMG = formula-fed group. Verbal Comprehension—VC, Perceptual Organization—PO, Freedom from Distractibility—FD, and Processing Speed—PS. a = HMG > MMG; b = FMG > MMG.

**Table 3 nutrients-17-03558-t003:** Delays in cognitive development frequencies according to neonatal feeding group.

	HMG (*n* = 20)	MMG (*n* = 22)	FMG (*n* = 20)	Total Sample (*n* = 62)	X2	*p*
Verbal IQ-VIQ	6 (30.0)	7 (31.8)	4 (20.0)	17 (27.4)	0.854	0.659
Performance IQ-PIQ	1 (5.0)	7 (31.8)	3 (15.0)	11 (17.7)	5.315	0.070
Total IQ	3 (15.0)	8 (36.4)	3 (15.0)	14 (22.6)	3.706	0.157
VC	3 (15.0)	7 (31.8)	2 (10.0)	12 (19.4)	3.554	0.169
PO	1 (5.0)	8 (36.4)	2 (10.0)	11 (17.7)	8.274	0.016 ^a^
FD	4 (20.0)	9 (40.9)	5 (25.0)	18 (29.0)	2.456	0.293
PS	3 (15.0)	8 (36.4)	5 (25.0)	16 (25.8)	2.507	0.285

NOTE. Values are *n* (%). HMG = human milk group; MMG = mixed milk group; FMG = formula-fed group. Verbal Comprehension—VC, Perceptual Organization—PO, Freedom from Distractibility—FD, and Processing Speed—PS. a = HMG > MMG.

**Table 4 nutrients-17-03558-t004:** Regression model identifying the significant predictors of WISC scores.

	R2Adj	F	*p*	B	SE	t	*p*
Verbal quotients	0.209	4.375	0.002				
Constant				96.388	13.994	6.888	<0.001
HMG and MMG vs. FMG				−0.010	0.010	−0.953	0.345
Maternal CBCL Internalizing symptoms				0.479	0.311	1.540	0.129
Maternal CBCL Externalizing symptoms				−0.976	0.315	−3.095	0.003
Paternal CBCL Internalizing symptoms				−0.461	0.322	−1.429	0.158
Paternal CBCL Externalizing symptoms				1.124	0.292	3.855	<0.001
Performance quotients	0.101	2.441	0.045				
Constant				103.767	10.980	9.451	<0.001
HMG and MMG vs. FMG				−0.013	0.008	−1.662	0.102
Maternal CBCL Internalizing symptoms				0.170	0.244	0.697	0.489
Maternal CBCL Externalizing symptoms				−0.533	0.247	−2.155	0.035
Paternal CBCL Internalizing symptoms				−0.157	0.253	−0.622	0.536
Paternal CBCL Externalizing symptoms				0.512	0.229	2.2.36	0.029
TOTAL IQ	0.196	4.124	0.003				
Constant				99.776	12.489	7.989	<0.001
HMG and MMG vs. FMG				−0.012	0.009	−1.365	0.177
Maternal CBCL Internalizing symptoms				0.371	0.277	1.337	0.186
Maternal CBCL Externalizing symptoms				−0.843	0.281	−2.994	0.004
Paternal CBCL Internalizing symptoms				−0.346	0.288	−1.202	0.234
Paternal CBCL Externalizing symptoms				0.907	0.260	3.484	0.001
Verbal Comprehension	0.161	3.460	0.008				
Constant				89.318	13.832	6.457	<0.001
HMG and MMG vs. FMG				−0.007	0.010	−0.704	0.484
Maternal CBCL Internalizing symptoms				0.435	0.307	1.415	0.162
Maternal CBCL Externalizing symptoms				−0.782	0.312	−2.508	0.015
Paternal CBCL Internalizing symptoms				−0.311	0.319	−0.977	0.333
Paternal CBCL Externalizing symptoms				0.957	0.532	3.321	0.002
Perceptual Organization	0.152	3.288	0.011				
Constant				105.119	11.922	8.818	<0.001
HMG and MMG vs. FMG				−0.019	−0.255	−2.188	0.033
Maternal CBCL Internalizing symptoms				0.366	0.289	1.384	0.172
Maternal CBCL Externalizing symptoms				−0.676	−0.366	−2.517	0.015
Paternal CBCL Internalizing symptoms				−0.340	−0.278	−1.236	0.221
Paternal CBCL Externalizing symptoms				0.672	0.435	2.703	0.009
Freedom from Distractibility	0.122	2.780	0.025				
Constant				121.860	15.017	8.115	<0.001
HMG and MMG vs. FMG				−0.012	0.011	−1.065	0.291
Maternal CBCL Internalizing symptoms				0.222	0.334	0.666	0.508
Maternal CBCL Externalizing symptoms				−0.908	0.338	−2.684	0.009
Paternal CBCL Internalizing symptoms				−0.545	0.346	−1.574	0.121
Paternal CBCL Externalizing symptoms				0.846	0.313	2.703	0.009

## Data Availability

The data presented in this study are available on request from the corresponding author due to privacy and ethical restrictions.

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
