# Peer review of "Long-Term Neurodevelopmental Outcomes of Late Preterm Children: A Pilot Study on the Role of Early Nutrition"

_nutrients, 2025, doi:10.3390/nu17223558_

Round 1
Reviewer 1 Report
Comments and Suggestions for Authors
Interesting idea of ​​this study, my recommendations are the following:
Abstract lines 27,29 I recommend that the name of the questionnaire be descriptive and only in parentheses acronyms.
I recommend that you also mention the specifics of the study subjects in the key words.
I recommend that in section 2.1. Sample the number of subjects included in the study be mentioned. I also recommend that you mention whether the questionnaire was completed by both parents or by a single random parent, or only the mothers.
I recommend that lines 132-135 be moved to section 2.1.
2.4.2. Cognitive development – ​​I recommend that you mention the way to answer this questionnaire.
2.4.3. Child psychological profile – I recommend that the reliability of this questionnaire be calculated taking into account the subjects' answers, by calculating the Cronhbach alpha index.
Table 1 recommends that all the acronyms mentioned below the table be descriptively mentioned, e.g. CC, APGAR.
3.1. Differences among Feeding groups on cognitive level of development – ​​I recommend that when interpreting the results, these should not be duplicated as information with those mentioned in table 1, I recommend revising.
Line 263, I recommend rewriting, a sentence cannot begin with – So,...
Discussion – I recommend expanding the section, by making new correlations between the results of the present study with results from previous studies. I recommend reorganizing this section by sentences, in a logic of ideas.
The Conclusions section is missing – I recommend mentioning it.
I also recommend mentioning future research directions at the end of the Discussion section.
I recommend that the bibliography be revised in accordance with the editing rules.
Author Response
We thank the reviewer for all the consideration and suggestions. A detailed reply is included in the attached file.

Reviewer 2 Report
Comments and Suggestions for Authors
Studying the long-term effects of nutrition in late preterm infants (8–12 years of age) is a rarely analyzed but clinically important issue. Attention should be paid to the role of fathers – a rare but valuable aspect that highlights differences in the perception of the child's problems between parents.
The relatively small sample size (n=62) limits statistical power, as reflected in the limitations of this study, but emphasizing this in the title would be justified – this is a pilot study.
The lack of a control group of full-term infants complicates the assessment of the relative level of deficits. If the authors could add such a group, it would be required.
There is a lack of data on the duration and continuity of breastfeeding and the age of weaning. Please provide a more detailed description of the study group selection criteria in section 2.4.1 (delivery method, antibiotic therapy, probiotic supplementation, and whether the pregnancy was terminated at term).
DOI: 10.3390/ijms26136424
There is a lack of information on the causes and mechanisms of the observed associations. The issue of maternal microbiota colonization during breastfeeding, the impact of the microbiome on metabolism, and, consequently, child development, should be clarified.
DOI: 10.3390/ijms26136424
Conducting studies on SCFA concentrations in the blood of the children studied, as metabolites of the gut microflora, could be a new direction for your research.
Potential interpretation errors – the positive correlation between fathers' high assessment of external problems and a higher IQ may be due to an artifact or differences in reporting methods. please explain
Author Response
We thank the reviewer for all the comments and suggestions that have helped us improve our manuscript. A detailed reply is included in the attached file.

Reviewer 3 Report
Comments and Suggestions for Authors
The review concerns the article entitled “Long-term neurodevelopmental outcomes of late preterm children: an exploratory study on the effects of early nutrition” presents interesting issues however some questions arise
The article analyzes the impact of early nutrition on the long-term cognitive development of late preterm children. The study includes 62 children aged 8–12 years and uses the Wechsler Intelligence Scale for Children, Third Edition (WISC-III) to assess cognitive development and the Child Behavior Checklist 6–18 (CBCL) to evaluate emotional and behavioral functioning.
The results suggest a positive effect of breastfeeding or formula feeding compared to mixed milk feeding, highlighting complex relationships between early nutrition and later cognitive outcomes
Study design and sample
- The small sample size (n=62) considerably limits the statistical power and generalizability of the findings. The study should be clearly described as pilot or exploratory. A sample size calculation should also be presented.
- The absence of a control group of full-term children makes it difficult to interpret group differences. This limitation should be explicitly mentioned in the Limitations of the Study section.
Methodology
- The Methods section lacks sufficient information on potential confounding variables (e.g., socioeconomic status, duration of breastfeeding, and age at weaning). Only the age, education level, and marital status of the mother and father are provided.
- The term mixed milk feeding is not operationally defined. The authors should specify the proportions of human and formula milk and the duration of feeding. This requires precise clarification.
- The regression analysis includes many predictors with a small sample, which increases the risk of overfitting; this should be discussed in the Discussion section.
- The recruitment process needs to be clarified — how were family pediatricians and children contacted? The number of families reached and the acceptance rate should be reported.
- Regarding the tools used (WISC-III and CBCL), please specify whether they were validated in the language of administration. Provide appropriate references and psychometric reliability values for each instrument.
- No information is provided about statistical power or correction for multiple comparisons.
- It would be valuable to include effect sizes (e.g., Cohen’s d) and confidence intervals.
- It is unclear whether the ANOVA tests met the assumptions of homogeneity of variance and normality.
Resutls
- The authors interpret group differences as an effect of feeding type, but other factors (e.g., maternal stress, home environment) may also contribute. The discussion should be more cautious and the interpretation refined.
- Findings should be formulated carefully, the data show correlations, not causation. This distinction is critical when presenting the findings.
- The discussion should emphasize the study’s limitations and avoid statements suggesting direct causal effects.
- Tables are extensive and difficult to read. Simplifying or aggregating WISC results and highlighting key group differences is recommended.
- The paper lacks visual representations (e.g., graphs) of the main relationships, which would aid interpretation. Consider adding these if possible.
Discussion
- Some paragraphs in the Discussion section repeat information from the Introduction. These should be shortened and clearly separated from the interpretation of findings.
- The discussion lacks a concise summary of the main clinical implications and directions for future research.
- The authors mention limitations but do not address missing data on feeding duration or the potential influence of COVID-19 on recruitment and results.
- A separate, structured subsection titled “Strengths and Limitations” should be added.
- A final Conclusion subsection is needed. In addition, the Conclusion part of the Abstract should be less general and more directly reflect the study results.
Author Response
We thank the reviewer for their comments and suggestions, which have helped us improve our manuscript. A detailed reply is included in the attached file.

Round 2
Reviewer 1 Report
Comments and Suggestions for Authors
No comments
Reviewer 2 Report
Comments and Suggestions for Authors
The authors responded to the suggestions, I have no further comments
best regards